# Meroterpenoids with Immunosuppressive Activity from Edible Fungus *Craterellus odoratus*

**DOI:** 10.3390/molecules28041564

**Published:** 2023-02-06

**Authors:** Quan Dai, Li-Ting Pang, Fa-Lei Zhang, Gang-Qiang Wang, Xue-Mei Li, Ji-Kai Liu, Tao Feng

**Affiliations:** 1School of Pharmaceutical Sciences, South-Central Minzu University, Wuhan 430074, China; 2Hubei Key Laboratory of Radiation Chemistry and Functional Materials, Non-Power Nuclear Technology Collaborative Innovation Center, School of Nuclear Technology and Chemistry and Biology, Hubei University of Science and Technology, Xianning 437100, China; 3Plant Protection and Quarantine Station of Dehong, Dehong 678400, China

**Keywords:** *Craterellus odoratus*, meroterpenoids, isolation and structure elucidation, immunosuppressive activity

## Abstract

Two unusual polyketide–sesquiterpene metabolites, craterodoratins T (**1**) and U (**2**), along with the known compound craterellin A (**3**), were isolated from the higher fungus *Craterellus odoratus*. The structures of isolated compounds were characterized based on nuclear magnetic resonance (NMR) and mass spectrum (MS) spectroscopic analysis, while the absolute configuration of the compounds was determined by theoretical NMR and electronic circular dichroism (ECD) calculations. Compound **1** possessed a rare structure with two aromatic groups. Compounds **1** and **3** showed immunosuppressive activity with IC_50_ values ranging from 5.516 to 19.953 *μ*M.

## 1. Introduction

Sesquiterpenoids are the most abundant among all natural products from fungi. Sesquiterpenoids are a class of terpenoids composed of a 15-carbon skeleton, which is prone to rearrangement with different skeletal structures, causing abundant structural variations leading to diverse bioactivities. Frequently, sesquiterpenes are moiety-coupled with non-terpenes to form meroterpenoids. Therefore, meroterpenoids have at least two substructures according to their definition: a terpenoid moiety and a non-terpenoid moiety. The non-terpenoid moiety can be derived from any biosynthetic pathway, but more than eighty percent of them are derived from a polyketide-derived pathway [1]. Polyketides from acyl-CoA thioesters catalyzed by polyketide synthase (PKS) are a large class of structurally diverse, acetate-derived natural products that exhibit a wide range of bioactivities. Drimane sesquiterpenes feature a 6/6-fused bicyclic carbon skeleton that is generally linked with a phenol or quinone/hydroquinone moiety to construct meroterpenoids. Best represented by puupehenone, avarol, and ilimaquinone [2,3,4], the sponge-derived merosesquiterpenes are a subgroup of meroterpenes, most of which possess either a drimane or a 4,9-friedodrimane terpenoid skeleton connected to a polyketide-derived hydroquinone or quinone. Several merosesquiterpenes are biologically active. For example, ilimaquinone [2] and epoxyphomalin A [5] are cytotoxic against various cancer cell lines, while ilimaquinone has shown other related activities, including anti-HIV and antituberculosis activities.

*Craterellus odoratus* is an edible mushroom (higher fungi) in the Cantharellaceae family. It is widespread in mainland China and characterized by possessing a bright orange or yellow cap [6,7,8]. Specifically, it has a funnel-shaped fragile basidiomata, hollow stipe, bright orange pileus, narrowly clavate basidia with narrower basidiospores, orange-yellow hymenial surface, clampless hyphae, and a strong pleasant odor [7]. Nutritional and biochemical analysis studies showed that a good amount of proteins, carbohydrates, reducing and non-reducing sugars, phenolics, flavonoids, β-carotene, and lycopene were found in the fresh fruiting bodies of *C. odoratus* [9]. Our previous studies on the secondary metabolites of *C. odoratus* have reported a series of polyketides, meroterpenes, and bergamotane-type sesquiterpenes [10,11,12,13,14]. Of them, craterellin B is a merosesquiterpenoid that showed significant inhibitory activity against the isozyme of human 11β-hydroxysteroid dehydrogenases (11β-HSD1), with an IC_50_ value of 3.5 *μ*g/mL [10], while craterellin C is also a merosesquiterpenoid from *C. odoratus* that showed inhibitory activities against two isozymes of human 11β-HSD1 and 11β-HSD2, with IC_50_ values of 14.8 and 25.4 *μ*g/mL, respectively [10]. Craterellin C also showed inhibitory activity against the human embryonic kidney 293 cell, with an IC_50_ value of 50.9 *μ*M [15]. In addition, a few bergamotane sesquiterpenoids from *C. odoratus* exhibited potent inhibitory activity against the lipopolysaccharide (LPS)-induced proliferation of B lymphocyte cells or against the concanavalin A (ConA)-induced proliferation of T lymphocyte cells [14]. These data suggest that *C. odoratus* is a good source of bioactive sesquiterpenes.

In order to search for more new and bioactive chemical constituents from this fungus, as well as to continue our ongoing search for structurally interesting and biologically active natural products from higher fungi, a continuous study on the cultures of *C. odoratus* in a rice medium has been carried out. As a result, two undescribed polyketide–sesquiterpene metabolites, craterodoratins T (**1**) and U (**2**), and one known compound, craterellin A (**3**) [10], were isolated from cultures of *C. odoratus* in the rice medium (Figure 1). Their structures with the absolute configurations were elucidated by extensive spectroscopic methods including 1D and 2D NMR, MS, UV, and IR technologies, as well as theoretical NMR and ECD calculations. All compounds were evaluated for their cytotoxic activities against five human cancer cell-lines, inhibitory abilities on nitric oxide (NO) production, and immunosuppressive activities. Herein, the isolation, structural elucidation, and bioactivities of these compounds are reported.

## 2. Results and Discussion

Compound **1** was isolated as a white powder. Its molecular formula (C_30_H_40_O_7_) was determined on the basis of the HR-ESIMS data at *m*/*z* 513.28479 [M + H]^+^ (calcd for C_30_H_41_O_7_^+^, 513.28468), corresponding to eleven degrees of unsaturation. The IR spectrum showed the presence of hydroxyl (3412 cm^−1^) and carbonyl groups (1681 cm^−1^). A detailed analysis of the ^1^H and ^13^C NMR spectra (Table 1) demonstrated the presence of five methyl signals, seven methylenes, eight methines (including three olefinic and three oxygenated), ten other carbons with no hydrogen attached, four carbon–carbon double bonds (*δ*_C_ 168.6, 163.0, 140.7, 136.1, 134.6, 123.2, 104.1, 99.1), one carboxyl group (*δ*_C_ 171.2), one α,β-unsaturated ketone (*δ*_C_ 197.1), and one epoxide (*δ*_C_ 58.0, *δ*_C_ 66.2), which accounted for seven degrees of unsaturation and four other degrees of unsaturation for four rings. Interpretation of the ^1^H-^1^H COSY, HMBC, and HSQC spectra allowed the deduction of the planar structure of compound **1**. The ^1^H-^1^H COSY spectrum suggested the presence of bold partial fragments (Figure 2), which were extended and connected to the skeletal framework by analyzing the HMBC correlation. Specifically, the HMBC correlations from *δ*_H_ 0.85 (3H, s, H-11) to C-3 (43.4), C-4 (33.9), C-5 (51.4), and C-12 (22.2); *δ*_H_ 0.78 (3H, s, H-13) to C-1 (40.6), C-10 (33.9) and C-9 (48.0); *δ*_H_ 1.62 (3H, s, H-14) to C-9 (48.0), C-8 (136.1) and C-7 (123.2); and from H_2_-15 to C-10 and C-9 demonstrated a sesquiterpene moiety, as shown in Figure 2 (red part). In the polyketide moiety, ring C was established by the HMBC correlations from H-2′ to C-3′ and C-4′; from H-5′ to C-3′, C-6′, and C-7′; and from H-6′ to C-2′, C-4′, and C-5′. The observed HMBC correlations from H_2_-15 to C-1′, C-2′, and C-6′ indicated that ring B was connected at C-1′ with ring C through C-15. The remaining C_9_H_11_O_4_ was verified by the HMBC correlations from H_3_-8″ to C-7″; from H-7″ to C-5″, C-6″, and C-8″; from H_2_-6″ to C-5″, C-7″, and C-8″; and from H-4″ to C-1″ and C-5″. The observed HMBC correlations from H_2_-7′ to C-1″, C-2″, and C-3″ indicated that ring C was connected at C-2″ with ring D through C-7′. Therefore, the planar structure of **1** was determined. A literature search suggested that the moiety of rings A–D was very similar to those of the potent cytotoxic fungal metabolites, peyronellins A and B [16]. The relative configuration of compound **1** was determined to be the same as that of peyronellin A [16] by the detailed analysis of the ROESY correlations between the resonances of H_3_-14 to H-1a and H-6b; H_3_-12 to H-1a and H-6b; H-5 to H-6a; and H-9 to H-1′ and H-1b, as shown in Figure 3. The stereochemistry of C-7″could not be elucidated by the ROESY data. To determine its final structure, the theoretical NMR calculations and DP4+ probability analyses were employed on two possible structures of (5*S^∗^*,9*S^∗^*,10*S^∗^*,1′*S^∗^*,2′*S^∗^*,6′*R^∗^*,7″*S^∗^*)-**1a** and (5*S^∗^*,9*S^∗^*,10*S^∗^*,1′*S^∗^*,2′*S^∗^*,6′*R^∗^*,7″*R^∗^*)-**1b**, and the calculations suggested that (5*S^∗^*,9*S^∗^*,10*S^∗^*,1′*S^∗^*,2′*S^∗^*,6′*R^∗^*,7″*R^∗^*)-**1b** was the correct relative configuration for **1** (see Appendix A). Both compound **1** and (5*S^∗^*,9*S^∗^*,10*S^∗^*,1′*S^∗^*,2′*S^∗^*,6′*R^∗^*,7″*R^∗^*)-**1b** had a negative Cotton effect at 300–350 nm in the ECD spectrum, while **1a** had a positive Cotton effect (Figure 4). So, the absolute configuration of **1** was established to be 5*S*,9*S*,10*S*,1′*S*,2′*S*,6′*R*,7″*R* by ECD calculations. Finally, the structure of compound **1** was established. It was named craterodoratin T after the name given in our previous work [14].

Compound **2** was isolated as a white powder. Its molecular formula of C_22_H_34_O_4_ was determined on the basis of the HR-ESIMS data at *m*/*z* 385.23468 [M + Na]^+^ (calcd for C_22_H_34_O_4_Na^+^, 385.23493), corresponding to six degrees of unsaturation. The IR spectrum showed the presence of hydroxyl groups (3360 cm^−1^). Comprehensive analysis of the NMR data (Table 1), including ^1^H- and ^13^C-NMR, indicated the presence of four methyls, five methylenes (one olefinic), eight methines (two olefinics and four hydroxylated carbons), and five other carbons without hydrogen attached, among which three were olefinic carbons. A detailed comparison of the NMR data between **1** and **2** showed that both had the same sesquiterpene moiety, as demonstrated by the COSY and HMBC correlations shown in Figure 2. Ring C was also determined to be present in **2** based on the HMBC correlations from the olefinic proton H-7′ to C-3′, C-4′, and C-5′, and from the oxygenated proton H-2′ to C-1′, C-3′, C-6′, and C-15, which indicated that ring C was connected at C-1′ with ring B through C-15. The ^1^H-^1^H COSY correlations of H-2′/H-3′ and H-5′/H-6′ suggested that the four oxygenated methine carbons are connected at C-2′, C-3′, C-5′ and C-6′, respectively. The relative configuration of the sesquiterpene moiety was presumed to be the same as that of **1** based on the ROESY correlations (H_3_-14/H-1a, H-6b, H_3_-12/H-6b, H_3_-11/H-5) (Figure 3). H-9/H-1b, H-5, and H-2′ were indicative of the relative configuration of the decalin portion of **2** as being 5*S**,9*S**,10*S**. The other four oxygenated methine carbons, according to the ROESY cross peaks of H-2′/H-9, H-3′/H-5′, and H-6′/H-5′, were established as 2′*S**, 3′*S**, 5′*R**, 6′*S** (Figure 3). The coupling constant of *J*_2′,3′_ = 3.2 Hz suggested that H-2′/H-3′ were on opposite sides, and *J*_5′,6′_ = 9.6 Hz suggested that H-5′/H-6′ were on the same side. Thus, the theoretical NMR calculations and DP4+ probability analyses were employed on two possible structures (see Appendix A). On the basis of these data, the absolute configuration of **2** was suggested to be 5*S*,9*S*,10*S*,2′*S*,3′*S*,5′*R*,6′*S* by the ECD calculations (Figure 4). Therefore, compound **2** was identified and named craterodoratin U.

The structure of compound **3** was identified as craterellin A (**3**) by comparison of its NMR spectroscopic data with those reported in the literature [10]. Compound **3** showed significant inhibitory activities against two isozymes of human 11β-hydroxysteroid dehydrogenases (11β-HSD1 and 11β-HSD2) with IC_50_ values of 9.1 and 1.5 *μ*g/mL, respectively [10]. In addition, compound **3** showed antibacterial activities against *Bacillus cereus*, *Escherichia coli*, *Staphylococcus aureus*, and *Micrococcus luteus*, with MIC values of 3.12, 6.25, 6.25, and 6.25 *μ*M, respectively [17]. These data suggest that craterellin A (**3**) has broad application prospects, including the treatment of obesity, diabetes, and bacterial infection. The total synthesis of 1′-*epi*-craterellin A from commercially available farnesol has been accomplished following a general strategy based on a sacrificial Diels–Alder–*retro*Diels–Alder approach to control regio- and stereoselectivity [18].

The simplest and most commonly seen reaction to generate fungal meroterpenoids is the introduction of a C_5_ isoprene unit by dimethylallyltryptophan synthase (DMATS) superfamily proteins. In this process, tryptophan and tryptophan-derived metabolites are accepted by DMATSs, which use dimethylallyl pyrophosphate (DMAPP) as a C_5_ prenyl donor [19]. In addition, the fungal meroterpenoids can be generated by one of two biosynthesis pathways: (i) the pathway in which a linear prenyl chain is attached to a non-terpenoid moiety and is cyclized later or remains uncyclized or (ii) the pathway in which the terpenoid moiety is already cyclized prior to being attached to the non-terpenoid moiety [19]. The final product is then released after modification by different post-modification enzymes.

The biosynthesis of the isolated compounds **1** and **2** is proposed as shown in Figure 1A. First, toluquinol (**A**) and compound **E** is derived from acetyl-CoA under the catalysis of PKS. Prenyltransferase catalyzes the coupling of toluquinol (**A**) with farnesyl pyrophosphate (FPP) to yield **B**, and then terpene cyclase (TC) cyclizes FPP into **C**. Compound **D** is obtained after a multi-step oxidation reaction. **D** can also be formed through the coupling between a cyclized sesquiterpene with an electron-rich vinyl moiety and an enone form of **A**, followed by epoxidation. Craterodoratin T (**1**) might be derived from **D** and **E**, and craterodoratin U (**2**) and craterellin A (**3**) might be derived from **D** via multiple steps of oxidation. Compound **1**, on the other hand, may originate from another pathway, as shown in Figure 1B. Coupling of FPP with **F,** which is derived from acetyl-CoA, can yield **G** and compound **1** can subsequently be produced by cyclization and epoxidation.

All compounds were evaluated for their cytotoxicity against five human cancer cell-lines using the MTT method. In addition, all compounds were evaluated for their anti-inflammatory activities by inhibiting NO production in LPS-activated RAW264.7 macrophages. However, all compounds were devoid of significant inhibitory activity at the concentration of 40 *μ*M in both cytotoxicity and anti-inflammatory assays. In addition, all compounds were further evaluated for immunosuppressive activity by inhibiting T- and B-lymphocyte proliferation. As shown in Table 2, compound **1** showed significant selective B-lymphocyte inhibitory activity, with relatively low cytotoxicity. Compound **3** also showed significant inhibitory activities against both T- and B-lymphocyte proliferation.

## 3. Materials and Methods

### 3.1. General Experimental Procedures

UV spectra were obtained by using a Double Beam Spectrophotometer UH5300 (Hitachi High-Technologies, Tokyo, Japan). IR spectra were obtained on a Shimadzu Fourier Transform Infrared spectrometer using KBr pellets. NMR spectra were recorded with a Bruker Avance III 600 MHz spectrometer (Bruker, Karlsruhe, Germany). Optical rotations were measured on a Rudolph Autopol IV polarimeter (Hackettstown, NJ, USA). HRESIMS were measured on an Agilent (Santa Clara, CA, USA) 6200 Q-TOF MS system. Circular dichroism (CD) spectra were measured with an Applied Photophysics spectrometer (Chirascan, New Haven, CT, USA). HPLC was performed on an Agilent 1260 liquid chromatography system equipped with Zorbax (Santa Clara, CA, USA) SB-C18 columns (5 *μ*m, 9.4 mm × 150 mm or 21.2 mm × 150 mm). Sephadex LH-20 (GE Healthcare, Chicago, IL, USA), Silica gel (200–300 mesh), and RP-18 gel (20–45 *μ*m, FuJi, Tokyo, Japan) were used for column chromatography (CC). Chiral separation was carried out on a Chiralpak AD-H chiral column (5 *μ*m, 250 × 4.6 mm; Daicel, Osaka, Japan).

### 3.2. Fungal Material

The fungus *Craterellus odoratus* (Schw.) Fr. was collected from the southern part of the Gaoligong Mountain in Yunnan Province, China, in July 2007. The fungus was identified by Prof. Mu Zang at the Kunming Institute of Botany. A voucher specimen (HFC2007-20180714-DQ1) was deposited in the School of Pharmaceutical Sciences, South-Central Minzu University.

### 3.3. Fermentation, Extraction, and Isolation

A piece of stored *C. odoratus* was transferred to potato dextrose broth (PDB) in a 500 mL Erlenmeyer flask and incubated at 24 °C until the mycelium biomass reached a maximum (normally in 6 days), which was used as the “seed” for large-scale fermentation in a rice medium. The seed was transferred to the rice medium and incubated at 24 °C in the dark for 40 days. The rice culture medium was composed of 5% glucose, 5% yeast, 0.15% pork peptone, 0.05% KH_2_PO_4_, and 0.05% MgSO_4_. The initial pH was adjusted to 6.0. A 250 mL Erlenmeyer flask containing 50 g of rice medium and 50 mL of water was sterilized at 121 °C for 15 min. A total of 400 flasks were used in this work.

The rice culture medium of *C. odoratus* (20 kg) was extracted six times with MeOH to produce an extract that was partitioned into water and ethyl acetate (EtOAc) layers. The EtOAc layer was concentrated under reduced pressure to give a crude extract (167 g), which was then subjected to CC over silica gel (200–300 mesh) and eluted with a solvent system of CHCl_3_–MeOH (1:0→0:1) to obtain nine fractions (A–I). Fraction D (20 g) was fractionated by MPLC over RP-18 silica gel column, then eluted with MeOH–H_2_O (from 5:95 to 100:0, *v*/*v*) to give 13 subfractions (D_1_–D_13_). Fraction D_5_ (3 g) was fractionated by CC over Sephadex LH-20 (MeOH) and then purified by prep-HPLC (CH_3_CN/H_2_O 28:72 in 35 min) to give **1** (3.0 mg, retention time (*t*_R_) = 21.3 min) and **2** (1.9 mg, *t*_R_ = 27.9 min). Fraction D_7_ (1 g) was purified by prep-HPLC (CH_3_CN/H_2_O 40:60 in 20 min) to give **3** (2.5 mg, *t*_R_ = 18.1 min).

*Craterodoratin T (**1**)*: white powder; [α]^21^_D_–8.0 (*c* 1.0, MeOH); UV (MeOH) *λ* _max_ (log *ε*) 215 (4.13), 285 (3.85) nm; IR (KBr) ν_max_ 3412, 2924, 1681, 1583, 1435, 1265 cm^−1^; ^13^C NMR data, see Table 1; ^1^H NMR data, see Table 2; HRESIMS *m*/*z* 513.28479 [M + H]^+^ (calcd for C_30_H_41_O_7_^+^, 513.28468).

*Craterodoratin U (**2**)*: white powder; [α]^21^_D_ + 15.2 (*c* 1.0, MeOH); UV (MeOH) *λ* _max_ (log *ε*) 210.0 (3.66) nm; IR (KBr) *ν*
_max_ 3360, 2945, 2833, 1454, 1112, 1031 cm^−1^; ^1^H and ^13^C NMR data, see Table 1; HRESIMS *m*/*z* 385.23468 [M + Na]^+^ (calcd for C_22_H_34_O_4_Na^+^, 385.23493).

### 3.4. NMR and ECD Calculations

#### 3.4.1. NMR Calculations

Conformation searches based on molecular mechanics with MMFF force fields were performed for stereoisomers to obtain stable conformers with populations higher than 1% [20,21]. All these conformers were further optimized according to the density functional theory method at the B3LYP/6-31G(d) level by the Gaussian 16 program package with g09 default keyword [22]. Gauge Independent Atomic Orbital (GIAO) calculations of their ^1^H and ^13^C NMR chemical shifts were conducted using density functional theory (DFT) at the mPW1PW91/6-311+G(d,p) level with the PCM model in methanol. The calculated NMR data of these conformers were averaged according to the Boltzmann distribution theory and their relative Gibbs free energy. The ^1^H and ^13^C NMR chemical shifts for TMS were also calculated by the same procedures and used as a reference. After calculation, the experimental and calculated data were evaluated by linear correlation coefficients (*R*^2^) and the improved-probability DP4^+^ method [23].

#### 3.4.2. ECD Calculations

All the stable conformers were further optimized by the density functional theory method at the B3LYP/6-31G(d) level by the Gaussian 16 program package with g09 default keyword. The ECD were calculated using density functional theory (TDDFT) at B3LYP/6-311+G(d,p) level in methanol with the IEFPCM model. The calculated ECD curves were all generated using SpecDis 1.71 program package, and the calculated ECD data of all conformers were Boltzmann-averaged by Gibbs free energy [24].

### 3.5. Cytotoxicity Assay

Five human cancer cell lines, breast cancer MCF-7, hepatocellular carcinoma SMMC-7721, human myeloid leukemia HL-60, colon cancer SW480, and lung cancer A-549 cells, were used in the cytotoxic assay. All the cells were cultured in RPMI-1640 or DMEM medium (HyClone, Logan, UT, USA), supplemented with 10% fetal bovine serum (HyClone, Logan, UT, USA) in 5% CO_2_ at 37 °C. The cytotoxicity assay was performed according to the MTT (3-(4,5-dimethylthiazol-2-yl)-2,5-diphenyl tetrazolium bromide) method in 96-well microplates [25]. Briefly, 100 *µ*L adherent cells were seeded into each well of 96-well cell culture plates and allowed to adhere for 12 h before drug addition, while suspended cells were seeded just before drug addition with an initial density of 1 × 10^5^ cells/mL. Each tumor cell line was exposed to the test compound at concentrations of 0.0625, 0.32, 1.6, 8, and 40 *μ*M in triplicate for 48 h, with taxol as a positive control. After compound treatment, cell viability was detected, and the cell growth curve was graphed. The IC_50_ value was calculated by Reed and Muench’s method [26].

### 3.6. Nitric Oxide Production in RAW 264.7 Macrophages

Murine monocytic RAW264.7 macrophages were dispensed into 96-well plates (2 × 10^5^ cells/well) containing RPMI-1640 medium (HyClone) with 10% FBS under a humidified atmosphere with 5% CO_2_ at 37 °C. After 24 h of preincubation, cells were treated with serial dilutions of the test compounds, up to a maximum concentration of 40 *μ*M, in the presence of 1 *μ*g/mL LPS for 18 h. The compounds were dissolved in DMSO and further diluted in the medium to produce different concentrations. NO production in each well was assessed by adding 100 *μ*L of Griess reagent (reagent A and reagent B, Sigma, St. Louis, MO, USA) to 100 *μ*L of each supernatant from the LPS-treated (Sigma) or LPS- and compound-treated cells in triplicate. After a 5 min incubation, the absorbance of samples was measured at 570 nm with a 2104 Envision multilabel plate reader (Perkin-Elmer Life Sciences, Inc., Boston, MA, USA). MG-132 was used as a positive control (IC_50_ = 0.2 *μ*M).

### 3.7. Immunosuppressive Activities Assay

#### 3.7.1. Preparation of Spleen Cells from Mice

Female BALB/c mice were sacrificed by cervical dislocation, and the spleens were removed aseptically. Mononuclear cell suspensions were prepared after cell debris and clumps were removed. Erythrocytes were depleted with ammonium chloride buffer solution. Lymphocytes were washed and resuspended in RPMI-1640 medium supplemented with 10% FBS, penicillin (100 U/mL), and streptomycin (100 mg/mL).

#### 3.7.2. Cytotoxicity Assay

Cytotoxicity was tested with Cell Counting Kit-8 (CCK-8) assay. Briefly, fresh spleen cells were gained from female BALB/c mice (18–20 g). Spleen cells (1 ×10^6^ cells) were seeded in triplicate in 96-well flat plates, then cultured at 37 °C for 48 h in 96-well flat plates in the presence or absence of various concentrations of compounds, as well as in a humidified and 5% CO_2_-containing incubator. A certain amount of CCK-8 was added to each well at the final 8–10 h of culture. At the end of culturing, we measured the OD values with a microplate reader (Bio-Rad 650, Hercules, CA, USA) at 450 nm. Cyclosporin A (CsA), an immunosuppressive agent, was used as a positive compound with definite activity, and the OD values from medium-only culture were used as background. The cytotoxicity of each compound was expressed as the concentration of compound that reduced cell viability to 50% (CC_50_).

#### 3.7.3. T-Cell and B-Cell Function Assay

Fresh spleen cells were obtained from female BALB/c mice (18–20 g). The 5 × 10^5^ spleen cells were cultured at the same conditions as those mentioned above. The cultures, in the presence or absence of various concentrations of compounds, were stimulated with 5 *µ*g/mL of concanavalin A (ConA) or 10 *µ*g/mL of lipopolysaccharide (LPS) to induce T-cells’ or B-cells’ proliferative responses, respectively. Proliferation was assessed in terms of uptake of [^3^H]-thymidine during 8 h of pulsing with 25 *µ*L/well of [^3^H]-thymidine, and then cells were harvested onto glass fiber filters. The incorporated radioactivity was counted using a Beta scintillation counter (MicroBeta Trilux; PerkinElmer Life Sciences, Waltham, MA, USA). Cells treated without any stimuli were used as a negative control. The immunosuppressive activity of each compound was expressed as the concentration of compound that inhibited ConA-induced T-cell proliferation or LPS-induced B-cell proliferation to 50% (IC_50_) of the control value. Both the cytotoxicity and proliferation assessment were repeated twice.

## 4. Conclusions

In summary, a total of three polyketide–sesquiterpene metabolites, including two new ones, were isolated from the edible fungus *C. odoratus*. Structurally, compound **1** is a novel merosesquiterpene that possesses a novel carbon skeleton with two aromatic groups. Compounds **1** and **3** exhibited selective inhibitions on ConA-induced T cell proliferation and LPS-induced B-cell proliferation. This study provides certain support for the further development and utilization of the edible mushroom *C. odoratus*.

## Data Availability

The data that support the findings of this study are available from the corresponding author (F.T.).

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
