# Peer review of "Meroterpenoids with Immunosuppressive Activity from Edible Fungus Craterellus odoratus"

_molecules, 2023, doi:10.3390/molecules28041564_

Round 1

Reviewer 1 Report

The manuscript “Meroterpenoids with immunosuppressive activity from edible 2 fungus Craterellus odoratus” is well written. The authors isolated two new compounds from C. odoratus and provided sufficient data and details to confirm them. This can be considered worth publishing in journal “Molecules” after following response and modifications:

1.  Line 28: References 1-3 are too old. It had better to replace them with some new ones.

2.  In the introduction section, the authors should introduce more information about C. Odoratus, such the main bioactivities of C. Odoratus? The main active compounds of C. Odoratus? Then, highlight the significance and value of this study.

3. As part of the results, Figure 1 appears in introduction section is a bit wacky, and the citation of Figure 1 is not marked in the introduction text.

4. Line 68-72: Could you provide the culture collection and number of the C. Odoratus strain?

5. Why the authors sterilized the medium only for 15 min (121oC)? Usually, the liquid medium should be sterilized for 15 min (121oC), and the solid medium should be sterilized for 30 min (121oC) or even longer. Thus, can the sterilization effect be guaranteed in 15 min?

6. Line 80: Is the “culture broth” solid or liquid state? How do the authors deal with them? Were the mycelia of C. Odoratus separated from the medium?

7. Is there any difference between craterellin A from C. Odoratus in structure and bioactivities from the known one?

8. Did the authors test the other bioactivies of the new compounds 1 and 2 except the immunosuppressive activity? Why?

9. A separate abbreviation should be provided.

10. There are still some format errors in the text and references, such as fonts, spaces, punctuation symbols, etc., please check the full text carefully.

Author Response

  1. Line 28: References 1-3 are too old. It had better to replace them with some new ones.

Re. We have updated the references.

  1. In the introduction section, the authors should introduce more information about C. Odoratus, such the main bioactivities of C. Odoratus? The main active compounds of C. Odoratus? Then, highlight the significance and value of this study.

Re. We have revised it.

  1. As part of the results, Figure 1 appears in introduction section is a bit wacky, and the citation of Figure 1 is not marked in the introduction text.

Re. We have revised it.

  1. Line 68-72: Could you provide the culture collection and number of the C. Odoratusstrain?

Re. The fungus was activated and passed on PDA medium and a voucher specimen (HFC2007-20180714-DQ1) has been deposited in the School of Pharmaceutical Sciences, South-Central Minzu University.

  1. Why the authors sterilized the medium only for 15 min (121oC)? Usually, the liquid medium should be sterilized for 15 min (121oC), and the solid medium should be sterilized for 30 min (121oC) or even longer. Thus, can the sterilization effect be guaranteed in 15 min?

Re. We have revised this section. In our experience, if the rice is cooked for too long, the rice will not form a relatively porous structure, which is conducive to the growth of mycelium. Of course, we must make sure that there won't be any contamination. 

  1. Line 80: Is the “culture broth” solid or liquid state? How do the authors deal with them? Were the mycelia of C. Odoratusseparated from the medium?

Re. We have revised this section. A “small” liquid culture broth of C. Odoratus was used as the “seeds” for large-scale fermentation in rice. The rice culture medium of C. odoratus (20 kg) was extracted six time with MeOH to give an extract that was partitioned into water and ethyl acetate (EtOAc) layers.

  1. Is there any difference between craterellin A from C. Odoratus in structure and bioactivities from the known one?

Re. The known compound, craterellin A (3), is consistent with that reported in the literature. And craterellin A was reported in the literature showed significant inhibitory activities against two isozymes of 11β-hydroxysteroid dehydrogenases (11β-HSD1 and 11β-HSD2) with IC50 value of 9.1 and 1.5 μg/mL, respectively.

  1. Did the authors test the other bioactivies of the new compounds 1 and 2 except the immunosuppressive activity? Why?

Re. In addition to immunosuppressive activity, we also tested their cytotoxicities against five human cancer cell lines using the MTT method and their inhibitory activities against NO production in LPS-activated RAW264.7 macrophages. But all compounds were devoid of inhibitory activity at the concentration of 40 μM, which were, therefore, not included in the article.

  1. A separate abbreviation should be provided.

Re. We have revised it.

  1. There are still some format errors in the text and references, such as fonts, spaces, punctuation symbols, etc., please check the full text carefully.

Re. We have checked the manuscript carefully, and made revisions.

Reviewer 2 Report

Dear author,

The article: "Meroterpenoids with immunosuppressive activity from edible fungus Craterellus odoratus is appropriate to publish in this journal if the author makes the following minor revisions:

1. Line 68: What does the "Fr" stand for? 

2. The experimental and calculated ECD spectra of compound 1 do not match well. The author should provide more discussion about this point.

3. Which electron acts as a nucleophile for structure B in Scheme 1 to generate structure C? Please check and write more clearly about this mechanism.

4. The NMR data of compound 1 showed d (0.9 Hz) at position 2; how does this proton couple with and show a J value of 0.9 Hz? 

5. Could you please double-check the J value of the proton position 15a (14.1) coupling with? 

Author Response

  1. Line 68: What does the "Fr" stand for? 

Re. 'Fr' is part of the Latin name (Craterellus odoratus (Schw.) Fr.) of the fungus.

  1. The experimental and calculated ECD spectra of compound 1 do not match well. The author should provide more discussion about this point.

Re. We have revised it.

  1. Which electron acts as a nucleophile for structure B in Scheme 1 to generate structure C? Please check and write more clearly about this mechanism.

Re. We have revised the Scheme 1.

  1. The NMR data of compound 1 showed d (0.9 Hz) at position 2; how does this proton couple with and show a J value of 0.9 Hz? 

 Re. We have revised. The δH 3.25 (1H, s, H-2′) at position 2′ is a singlet.

  1. Could you please double-check the J value of the proton position 15a (14.1) coupling with? 

Re. We have revised. The 2J value of the proton H-15a (14.1) coupling with H-15b is 14.1 Hz.

Round 2

Reviewer 1 Report

The manuscript was revised accordingly.